# Densification of Bamboo: State of the Art

**DOI:** 10.3390/ma13194346

**Published:** 2020-09-29

**Authors:** Marzieh Kadivar, Christian Gauss, Khosrow Ghavami, Holmer Savastano

**Affiliations:** 1Research Nucleus on Materials for Biosystems (NAP BioSMat), Department of Biosystems Engineering, University of São Paulo; Pirassununga, Sao Paolo 13635-900, Brazil; cgauss@waikato.ac.nz (C.G.); holmersj@usp.br (H.S.J.); 2School of Engineering, University of Waikato, Hamilton 3216, New Zealand; 3Department of Civil Engineering, Pontifical Catholic University, Rio de Janeiro 22453-900, Brazil; ghavami@puc-rio.br

**Keywords:** bamboo, densification, thermo-mechanical, viscoelastic-thermal-compression

## Abstract

Densification processes are used to improve the mechanical and physical properties of lignocellulose materials by either collapsing the cell cavities or by filling up the pores, consequently reducing the void volume fraction. This paper focuses on an extensive review of bamboo densification process, which is achieved by compressing the material in the direction perpendicular to the fibers using mainly two different techniques: an open system, thermo-mechanical (TM), or a closed system, viscoelastic-thermal-compression (VTC). The main aim of bamboo densification is to decrease its heterogeneity, as well as to improve its mechanical and physical performance. In addition, densification may occur during the manufacturing of bamboo products in which hot-pressing processes are used to mold bamboo panels. There are over 1600 publications about bamboo, concentrated in the recent decade, mainly about engineered materials. Although several papers regarding bamboo and wood densification are available, very few studies have comprehensively investigated the densification process solely through compression of natural bamboo culms. According to the literature, applying a combination of compression of 6–12 MPa at temperatures between 120–170 °C for 8–20 min can produce materials with higher strength in comparison to the mechanical properties of natural bamboo. The majority of research on bamboo densification indicates that the modified material results in improved properties in terms of density, hardness, bending strength, stiffness, and durability. This paper provides a review that consolidates knowledge on the concept of bamboo culm densification, discusses the roles of parameters that control the process, ascertains the best practice, and finally determines gaps in this field of knowledge.

## 1. Introduction

Sustainable materials are in high demand, particularly within the forest products industry, due to the increased need to decarbonize the built environment. Population and economic growth are straining the finite supply of suitable products. In addition, climatic changes and environmental impacts of industrial products are leading to the development of eco-friendly resources [1]. According to life cycle assessment studies and considering the three pillars of sustainability (e.g., environmental, social, and economic impacts), bamboo has indisputable potential as a sustainable resource for a wide range of utilizations [2,3,4,5,6,7]. Bamboo as a raw material has a beautiful aesthetic, good mechanical strength, and is harvested in short rotation periods; however, it also has disadvantages [8]. Variable cylindrical geometry, heterogeneity, and the variability of properties of full-culm bamboo generate challenges for its use in mainstream production. This has led to engineering the material to obtain standardized prismatic shapes and less variability in mechanical properties and performance. Consequently, a list of bamboo-based panels (BBP) having a large flat surface and different bamboo units have been designed and produced to overcome these disadvantages since the 1970s [8]. Mechanical, physical, and aesthetical properties of BBPs can be controlled and engineered using specific species and processing methodologies. Although most methods follow the advancement of modern timber industry [9], there are differences in the bamboo-based production due to the geometry and the material consistency.

Densification is used to manufacture bamboo scrimber and could also be used in the production of other bamboo products. Increasing the density of bamboo elements can lead to more efficient use of structural products because the quality of the final product depends on the performance of its components. Wood densification has been investigated and comprehensively reported [10,11,12,13,14]. Kutnar et al. 2015 summarized the state of the art and knowledge in the field of compressed wood processing and products [14]. Notwithstanding, there is still a lack of knowledge about the fundamental features of bamboo full-culm densification, particularly the softening behavior, physical properties, and effective processing parameters [15]. Fundamental understanding of the bamboo element structure and engineering properties, as well as bamboo processing, can open up new advanced methodologies to promote its use for new applications [16]. In this respect, flattening and densification are two new technologies that have been explored in the market and have recently been published in the academic literature. Flattening can overcome the circular shape of bamboo and densification makes it homogeneous across the wall thickness. A combination of these two technologies makes it possible for inhomogeneous round shape bamboo to be substituted by a uniform flat material so that commercially uninteresting features of bamboo could be altered. The objective of this paper is to present the state of the art of densified bamboo and provide a review of the publications related to the effective factors for the bamboo densification process.

## 2. Densification Concept

Densification can be defined as a process to increase the density and redesign the microstructure of the material. Chemical, thermal, or mechanical factors can influence this process. In the case of materials with cellular structure, densification has been defined as the last regime of stress-strain curves in compression after linear elastic and plateau region [17]. In these types of materials like wood, the conventional understanding of the stress-strain relationship under compression identifies three distinct regions. In the beginning, the material exhibits elastic behavior, where load and deformation are linearly related. At the end of the elastic regime, the cells collapse and an inelastic behavior commences. In the inelastic region, strain increases rapidly with little or no change in stress, often called the plateau region. If compressive loading continues, all cells will collapse and the cell cavities are removed, transforming the material to function as a solid body, which will dramatically increase the stress. This region is called densification [17,18,19,20]. The domain of these regions and their starting and ending points depend on the type and microstructural features of the material. 

The mechanical behavior of a cellular structure, like bamboo and wood, can be modelled by different methods during a compressive force. The unit cell method considers the geometry of each cell structure in two or three dimensions. The method of dimensional analysis, on the other hand, relies on the correlation between the relative density and mechanical properties, which can be an easier and more accurate approach [17,21]. Simulation of the material using finite element analysis (FEA) is also another method, which, considering the geometry and local effects, can be very accurate, and in some cases requires intense computations [17].

On the other hand, impregnating the material with additives is the other mechanism to achieve a greater weight and thereupon density. It is possible to fill the lumens and pores of a lignocellulose material with a suitable substance such as resins. This process causes the formation of new composite material with different physical, chemical, and mechanical properties. In some cases, both mechanisms may occur at the same time in some processes, like in the production of compressed wood-polymer composites [22] and bamboo scrimber [23,24,25]. The final density is controlled by the densification method. In addition, the physical, mechanical, and chemical properties of the raw materials are the key features. Porosity, for example, greatly affects the subsequent processing and mechanical properties of the final product.

This paper is focused on the first densification mechanism in which the volume of the material is changed using compression at elevated temperatures. The increase in the density of wood by compression at high temperatures has been done for at least a century, exploring different wood species [26]. It worth mentioning that for lignocellulose materials like wood, when the utilized procedure involves high temperatures, not only does the wood structure change, but chemical modifications may also occur during the process, which affects wood properties. Bamboo, as a lignocellulosic material, follows the same model. However, due to the lack of sufficient data in this field, there is a gap in understanding the factors and the impact on the densification process. Further research is needed to quantify and predict the behavior of bamboo during the densification process accurately.

## 3. Bamboo

Bamboos are fast-growing woody grass plants that are subdivided into different families addressed in several reports [27,28], with more than 1200 species found globally. It is composed of the rhizome, which is buried underground, and the culm, which grows aboveground [27]. The part that is being used in the industry for the engineered bamboo production is the culm, which contains the woody material, formed with nodes and internodes. The internodes have a hollow cylindrical shape, while the nodes are thin diaphragms separating the internodes. The geometry of the culm is defined by the diameter, wall thickness, and internode length, which vary along with the height. Chaowana et al. (2017) [29] showed this transfiguration of the bamboo culm and alteration of the macroscopic characteristics for four different bamboo species. From their results, it can be inferred that the middle part of a bamboo culm is more homogeneous than the upper and bottom parts (Figure 1). The difference in the structure of these three parts, top, middle, and bottom, leads to dissimilarity in physical and mechanical properties. Thus, bamboo manufacturing primarily utilizes the bottom and middle thirds of a culm.

Anatomically, bamboo consists of fiber bundles (40%), parenchymal cells (50%), and vessels (10%), as shown in Figure 2 [28,30,31]. The mechanical properties of bamboo are correlated to its structure and fiber orientation. The fibers reinforce and support the matrix, which is composed of parenchyma. Therefore, at a macro scale, it is possible to consider bamboo as a uni-directional natural fiber reinforced composite. In contrast, at the meso-level, it can be seen through the bamboo culm wall thickness, as shown in Figure 3, that the fiber distribution is not homogenous. The density of fibers is functionally graded, increasing from inner to the outer region of the wall [30,32,33]. In addition, the fiber concentration is higher in the upper half of the culm than the bottom [28,34]. The lack of homogeneity across the bamboo thickness is one of the negative points of using bamboo culm directly in the construction industry.

The oven-dried density of bamboo is around 400–900 kg/m^3^ [35] and varies by the species, age, growing conditions, and location along the length of the pole. There are many cavities within the bamboo vascular bundles, as presented in Figure 4. In addition, in bamboo fibers, which are bundles of elementary filaments bonded together by the middle lamellae [36] (Figure 5), the proportion of lumen tends to be higher towards the fiber bundle periphery. Therefore, theoretically, it is possible to compress bamboo close to a density of the cell wall density, which is approximately 1500 kg/m^3^ [37]. This compression could be in the radial or tangential direction to close the vessels, and the required force depends on the strength of the cell walls to buckle and varies by the species.

These features of bamboo, as well as the variability of the geometric properties, have prevented the industry from considering raw bamboo as a mainstream building material. To address these limitations, the industry has shifted to manufacturing bamboo-based panels that are comparable to wood and engineered wood board products.

## 4. Densification in Bamboo-Based Panels

Bamboo based panels (BBPs) are widely applied in the field of construction, in surface applications (e.g., floors, ceilings, and wall finishes), as well as used in the furniture industry. BBPs are categorized and nominated based on the terminology standard. Referring to the Chinese standard “LY/T 1660-2006 Standard terminology for bamboo-based panels” [38], Liu et al. (2016) [39], and Huang (2019) [40], standard BBP products are labeled as follows:Flattened bamboo panel, formed by flattening the bamboo culm [41].Bamboo laminated lumber, constituted of bamboo strips [42].Plybamboo, made of bamboo slivers, or bamboo mats, and curtains obtained by weaving [43].Bamboo scrimber, by gluing the bamboo fiber bundle [44].Bamboo particleboard, by molding the fine bamboo particles [45,46].Bamboo oriented strand board, by molding the oriented strand elements [47,48,49].

In the classified products, the flattened board is obtained from circular bamboo culm without using resin, while the others are developed by a “decomposition” and “recombination” process [8]. The production processes of these bamboo-based panels are summarized in Figure 6. During decomposition, the bamboo culm is broken down into small segments (engineered bamboo elements) such as veneers, strips, slivers, strands, and particles. Through the use of resin and hot press, it is possible to recombine the small elements and produce a board.

It is possible to classify the bamboo elements required for the six mentioned bamboo-based panel products based on the geometry of the elements. This classification divides bamboo units into six major groups (Table 1). As shown in Figure 6, hot-pressing is a significant step for the manufacture of bamboo products in which bamboo elements bond together and might undergo compaction and increase their density. Among the BBPs mentioned, densification is more likely to occur in the flattened bamboo panels and bamboo scrimber, which alter the cell structures and improve the weak tissues.

### 4.1. Densification in Flattened Bamboo Panels

In the available literature on bamboo, there are generally three concepts of bamboo geometry modification: reformed bamboo, flattening, and densification. Reformed bamboo is a term which only exists in the literature from 1990s until 2003 and was a combination of flattening and densification [31,59,60]. The concept of reformed bamboo changed after 2003 and divided into two different mechanisms of deformation: flattening and densification [15,16,31,41,59,60,61,62,63,64,65,66,67,68]. Bamboo flattening is a technique that softens and flattens the tubular bamboo culm directly into a flat board and to achieve that several different processes and types of equipment have been tested [41,61,62,63]. The method has some disadvantages, including the development of deep cracks during flattening. The mechanism of flattening, which is presented in Figure 7, occurs with the extension of the inner zone (tension) and contraction (compression) of the outer zone, along the tangential direction.

Fang et al. (2018), [41] reviewed the flattening methods and suggested two possible solutions to prevent the development of cracks. The first suggested solution is reducing the circumference differentiation of the internal and external layers, which is possible by extracting the skin or splitting the culm into several pieces or executing superficial scratches on the inner surface. The second method is to decrease the tangential stress by chemical treatment, heat treatment, or/and increasing moisture content. The combination of these two solutions, modification of the geometry and material treatment, can be more effective. Employing microwave heating [69] using chemical reagents [70], hot oil immersion [71], and applying high-pressure steam [63,72] are successful approaches for softening the half-split bamboo culm and flattening to non-cracked bamboo board. There are also several investigations concentrated on the technological factors and tools to facilitate the flattening process by counteracting the stress of the bamboo material, mainly focused on the internal culm wall during deformation [61].

Densification is one of the improvements that has been suggested as a post-treatment of flattened bamboo culms to maintain the new geometry and to reduce indentations. In the case of using the method of superficial scratches on the inner layer of bamboo, after just the flattening process, the small cuts will open (Figure 8) because of tension in this region. Therefore, densification will be required to fill up the grooves.

### 4.2. Densification in BAMBOO Scrimber

Bamboo scrimber, also known as reconstituted densified bamboo [73], utilizes fiber bundles dipped in phenol-formaldehyde resin, which is then compressed in a mold and heated to cross-link the resin [74]. Due to its excellent physical and mechanical performance compared to natural bamboo [53,75,76], scrimber has attracted the attention of the bamboo industry in the 2000s [8]. However, the quality of this engineered bamboo product depends on the bamboo species, media treatment, type and content of the resin, as well as the pressing parameters [23,77,78,79], which are also related to the density. The density of a bamboo scrimber can be as high as 1050–1250 kg/m^3^ [73], which is almost twice the density of raw bamboo. Physical and mechanical properties of scrimber depend on the density. The higher the density, the higher the mechanical properties, and the lower the water absorption [73,79,80].

The increase of density may be related to the closure of bamboo voids such as vessels, parenchymas, and fiber lumens caused by high-pressure and hot-pressing processes [25,75], or the presence of solid resin particles in the cavities. The microstructural analysis of engineered bamboo scrimber at different densities conducted by Yu (2017) [75], shows that when the scrimber density is low, for example at 850 kg/m^3^, the structure is irregular, which led to deficient fiber bonding. Some of the big vessels are open, and just a few parenchyma cells are compacted. By increasing the density of bamboo scrimber to 1300 kg/m^3^, the structure gets more compact and the fibers bond more effectively. On the other hand, part of this increase in density is due to the addition of the resin.

## 5. Densified Bamboo

Densified bamboo utilizes the full-culm (or strips with whole thickness of bamboo) with the fiber bundles maintained in the parenchyma matrix, which is then compressed to densify the culm wall. This product can be used as a single lamina or be laminated into a multiply board with or without adhesive. The main aim of the bamboo densification is to decrease its heterogeneity and improve its mechanical properties since, in raw bamboo, most of the characteristics change through the bamboo transversal section. The densification process can change the distribution of fibers through the bamboo cross-section and, consequently, homogenize the distribution of physical and mechanical properties. The method can be achieved by pressing bamboo perpendicular to fibers, which can be transversal or radial direction (the directions are shown in Figure 9). Flattening bamboo results in compression in both transverse and radial directions. However, in this study, the densification of the culm wall is only considered in the radial direction.

To densify bamboo in the radial direction, the applied force should overcome the compressive strength of bamboo to buckle the cells and pass the elastic region. Typical compressive stress-strain curves of *Phyllostachys edulis* (Moso) bamboo are shown in Figure 9 (red line). Although the compression test had been carried out at room temperature, it is still possible to understand that using more than 30 MPa of pressure can cause a deformation higher than 40%. Elevated temperatures and using the same pressure can lead to a higher densification degree (DD). To facilitate bamboo deformation (for both flattening and densification mechanisms), its viscoelastic behavior must be considered. Similar to other viscoelastic materials, temperature and glass transition temperature (Tg) are the most significant parameters for geometric deformation.

### 5.1. Softening Behavior of Bamboo

Bamboo contains natural polymers based on hemicellulose, lignin, and non-crystalline cellulose components, which are amorphous phases. Therefore, it is considered a viscoelastic material [81,82,83]. The mechanical behavior of bamboo is between linear elastic solids and viscous fluids, similar to other viscoelastic materials such as polymers and wood. Figure 10 presents the different stages of amorphous polymers and the resulting behavior with increasing temperature. In this figure, the Tg limit, or the softening temperature stage, describes the softening characteristics of amorphous polymers in which many qualities of the material change sharply. For example, the molecular movement and damping properties increase while the strength and elastic modulus fall dramatically [84,85,86,87]. At temperatures below Tg, bamboo behavior is glassy, while at higher temperatures, it behaves like a rubbery or viscous material [88].

The glass transition in these viscoelastic materials is dependent on the moisture content, chemical composition, and testing method. For wood, as an example, the Tg ranges from 60 °C to 235 °C [88]. Higher moisture contents lead to a decrease in the glass transition temperature. As presented in Figure 10, the increase in temperature makes the amorphous material constituents to display viscous behavior. However, this stage is not reached in wood before the start of thermal degradation, and the amorphous wood components will degrade at elevated temperatures [82,84]. Bamboo also experiences similar stages as wood when exposed to elevated temperatures, and the mechanical properties are also dependent on temperature and moisture.

In relative terms, at short durations, low temperatures, and low moisture content, amorphous bamboo constituents stay in a “glassy state” and exhibit high strength and modulus. By increasing the temperature, moisture content, and duration, bamboo exhibits a rubbery behavior, which makes it easier to be deformed. Therefore, the softening stage needs to be considered in any geometry deformation mechanism of bamboo. Matan et al. (2007) [82] showed the glass transition temperature dependence on the bamboo’s initial moisture content for *Dendrocalamus asper* bamboo (Figure 11). According to their study, the Tg of bamboo approaches a constant value with initial moisture contents above 13% (between 100 and 120 °C) [82].

### 5.2. Densification Methods

Selection of the right method for densification requires accurate knowledge of the material, such as its anatomical characteristics, softening behavior, and also loading direction. Applying an improper method, using solely compression, for example, can cause problems in the final product. Spring back effect, which occurs when the sample does not maintain the targeted thickness after decompression, is one of the main problems associated with the densification process. It can be eliminated by using an appropriate method. An appropriate method is a process in which the optimized parameters apply to the material. Several methods that utilize steaming or heating, which can induce permanent fixation of the compressive deformation, have been used in the wood and bamboo industry [10,11,16,63,89,90,91]. The best densification method is the one in which plasticization of the material and stabilization of the final product is adequately taken into account.

There are approximately 1000 patents on bamboo densification, using different methods, mostly from China and Japan [92,93]. However, less than ten published studies of bamboo culm densification are published to date (presented in Table 2 and Table 3). These studies focused on thermo-mechanical (TM) method, which is conducted in an open system using heat and pressure, and viscoelastic-thermal-compression (VTC) process approach, using heat and pressure and pre-softening with steam in a closed system.

#### 5.2.1. Thermo-Mechanical (TM)

In thermo-mechanical (TM) densification, an open system with temperature and pressure is used to densify the material. TM operation processes have been applied between 140–200 °C, at 40%, 50%, and 60% compression rates using different moisture contents [11]. TM densification is usually performed in open systems (e.g., hot-press) in which controlling the sample moisture content (MC) or the relative humidity (RH) of the environment during the process is not possible. Therefore, in this method, special attention should be paid to the initial MC; low MC makes it difficult for the material to be densified, and high MC can cause an explosion. Stabilizing wood at 13% MC [100], and bamboo at 10% [65] is suggested to be enough to facilitate the process. Some commercial products developed during the 1950s were produced through this method [101].

Research has shown that both wood and bamboo materials exposed to TM densification have higher bending resistance and modulus of elasticity than the natural one [65,102]. The influence of effective parameters involved in TM process on the final densified material has been studied [103,104]. The reason for increasing the strength after TM densification process is related to the decrease in porosity and the increase in density [90]. However, at low temperatures and high compression rates, cellular cracks occur more often, which may decrease strength [102].

#### 5.2.2. Viscoelastic-Thermal-Compression

Densification of bamboo by viscoelastic-thermal-compression (VTC) is similar to that of wood, and includes the following manufacturing steps [11]:Elevating the temperature to exceed its glass transition temperature.Causing rapid steam decompression and removal of the bound water in the cell wall.Densification by compressing the material perpendicular to the grain.Relaxation of the remaining stresses. This step promotes the thermal degradation of the hemicelluloses in the material component.Cooling the material by conditioning that to the ambient temperature and humidity.

The critical point in the VTC process is softening in a high-pressure vapor environment. Yet, the material must have some initial moisture content, which changes during the pressing process due to the steam pressure in the closed space. For example, the initial moisture of wood to be VTC processed is preferable to be around 15–30% [105]. The desirable temperature and pressure levels depend on the initial density and species of the wood [105,106]. For low-density woods, the best temperature is between 160–175 °C and pressures of 650–2000 kPa [11]. However, for a higher density, a temperature of 175–225 °C with pressures of 2000–4000 kPa are required [11]. In the case of VTC process for bamboo, because of lack of information, the process parameters are not optimized yet. According to the available literature of wood and bamboo, the VTC densification method improves the mechanical properties of the material and enables dimensional stabilization [105,107,108].

#### 5.2.3. Mechanisms of Deformation

The deformation in the reviewed densification methods is controlled by three mechanisms:Purely mechanical densification, mainly from rearrangement of fiber bundles, brittle crushing, cell wall buckling, and subsequent moisture content drainage during the compaction.Materials shrinkage and plastic yielding due to elevated temperature (at a temperature higher than 160 °C, the chemical changes also cause additional deformation).Compaction resulting from intercell steam pressure, generally induced by stress and heat, which causes intercell cracks (even in TM method, the moisture content and elevated temperature can generate steam).

Therefore, it is possible to consider the VTC and TM as two types of THM (thermo-hydro-mechanical method), which are conducted in closed and open systems, respectively. There are other new densification methods such as thermo-vibro-mechanic (TVM), which is a new application using heat pressure and vibration [11]. However, very few studies applied this method to densify wood, and in the case of bamboo, there is no published literature at the time of writing.

## 6. Mechanical and Physical Performance of Densified Bamboo

The mechanical and physical qualities of densified bamboo subjected to the process parameters presented in Table 2 are summarized in Table 3. In general, three-point bending (B) and tensile tests (T) are used to evaluate the mechanical performance of densified bamboo. Based on the results in Table 3, without considering the effect of different species for both the natural and densified bamboo using different processing methods, the bending modulus of rupture (MOR) and modulus of elasticity (MOE) appear to be linearly correlated to the density (Figure 12). Dixon et al. (2016) [16] also showed the linear relationship of bending and density for natural and densified bamboo. The authors evaluated the bending characteristics in the longitudinal orientation for un-densified and THM densified bamboo (*Moso species*) and concluded the increase of MOE and MOR with densification. However, comparing the un-densified and densified bamboo with equal density, raw bamboo shows higher strength [16]. Due to the small number of specimens tested in tension, it is not possible to indicate any correlation for the tensile strength of densified bamboo.

## 7. Discussion of the Effective Parameters

### 7.1. The Influence of Bamboo Species

Bamboo species have varying physical and mechanical properties, related to the wall thickness, density, and fiber distribution. Accordingly, the densification mechanism varies by species. Tanaka et al. (2006) [68] adopted the densification method, which is presented in Figure 13, to produce the bamboo connector (for developing a new connecting system) using two different bamboo species, Moso and *Phyllostachys bambusoides* (Madake). To densify the material, the authors compressed sections of the bamboo culm wall using the press parameters shown in Table 2. As a result of this process, the density, tensile strength, and modulus increased. Although the values of tensile strength for the two bamboo species improved significantly by the densification process, the relative improvement was different. By using the same procedure, Moso tensile strength improved by 287.5% and reached 310 MPa. However, Madake tensile strength changed only by 29.41% from 170 MPa to 220 MPa [68]. The initial density of the natural Madake is less than that of the natural Moso, and it is expected to result in a higher density after the same pressure. However, different initial compression strengths not explored in the study may be the cause of the lower density for Madake bamboo. Different fiber content can be a justification to explain the different relative improvement of these two bamboo species.

### 7.2. The Influence of Pressing Temperature

As discussed in Section 5.1, an elevated temperature is needed in the bamboo densification process to facilitate deformation without generating a high incidence of cracks. There is the question of how much temperature is required and what the appropriate temperature range is for the bamboo press process. The answer to this question requires understanding the effect of temperature on bamboo material. The temperature must be high enough to make the bamboo plastic, but not too much to reduce mechanical strength and cause thermal degradation.

To evaluate the impact of pressing temperature on the mechanical strength of densified bamboo, Takagi et al. (2008) [67] prepared bamboo strips without nodes and placed them in an environment of 5 °C after soaking in water. Then, they dried samples at 50 °C for 22 h prior to the process to regulate the MC of samples. The authors used hot pressing at various temperatures of up to 220 °C for enhancing the flexural, compression, and impact resistance of bamboo. According to their results, the optimum process temperature was 220 °C in terms of obtaining the best density and flexural modulus, while 160 °C identified to be the optimized temperature in terms of flexural strength (as shown in Figure 14) [67].

### 7.3. The Influence of Densification Degree

THM pressure is one of the most significant determinants for the densification of bamboo. Semple et al. (2013) [94] clearly showed this effectiveness by densifying small specimens of Moso bamboo strips to produce samples with four different densification degrees using the THM method. Based on the study, the higher the compression ratio, the higher the density and MOR. By compressing the bamboo to 50% of the original thickness (50% densification degree), the method increased the MOR by 71% and the MOE by 31% (Table 3). However, strips compressed to 50% of the initial thickness generated a partial side spread of the tissue, and lateral expansion happens. Compressing to 33% (67% densification degree) causes notable tissue displacement and distortion. Figure 15 shows similar behavior for bamboo *D.Asper*. Therefore, a densification degree (DD) around 50% has been suggested to be the optimum compression ratio (CR) for the elimination of collapsible void space.

### 7.4. The Influence of Water and Initial Moisture Content

In Figure 11, the graph demonstrates that the softening behavior depends on initial moisture content. Water can facilitate bamboo softening and consequently the densification because of material plasticization [64,82]. Santos et al. (2014) [64] immersed *Guadua angustifolia Kunth* in water and densified it using thermo-hydro-mechanical (THM) treatments. The authors examined this pre-treatment by evaluating the tensile properties of THM modified *Guadua* and comparing it with the tension strength of undensified samples. The results showed the improvement of specific stiffness of the material by a factor of 1.25 as the result of pre-soaking the specimens.

Kadivar et al. (2019) [65] used an open thermal press to densify bamboo *D. asper* in its radial direction using various initial MC, from 0 to 20%. The study evaluated the effect of starting MC on bending and physical-chemical characteristics of the material. Applying the TM parameters presented in Table 2, the material achieved a maximum of 31.2% DD. According to the results, the initial MC is one of the effective parameters that needs to be considered for bamboo densification. Bamboo samples with less than 5% MC manifested cracks in TM processing, and consequently, the final product does not perform well in the presence of water. A high MC prevents uniform densification (Figure 16), and in this case, the samples need more time to release internal gas pressure. Based on the results, initial moisture content of approximately 10% was observed to be optimal, which can be high enough to satisfy the softening requirements for TM process and homogeneity in the densified product. Using this optimized MC, the densified bamboo achieved a 56% increase in ultimate stress in bending (Figure 17). However, in terms of physical properties and dimensional stability, the densified samples in all samples resulted in decreased performance compared to the natural bamboo.

### 7.5. The Microstructure of Densified Bamboo

Li et al. (1994) [31] produced reformed bamboo using a process including softening, followed by compressing and fixing the material and studied its microstructure. The study included optical microscopy of bamboo cross-sections before and after the process which revealed the rearrangement of fibers, dense compaction of vascular bundles, disappearance, and closure of cavities, as well as deformation of the components configuration. The authors used an Automatic Image Analyzer for estimating the ratio of fibers to the total area, and their fiber volume fraction (Vf) (Figure 18). For the natural bamboo, Vf gradient declines along the thickness. After the reforming process, the Vf is uniformly distributed, about 50% through the thickness. However, adjacent to the inner layer, the fiber distribution appears not to change due to the process [31].

Several researchers have reported the microstructure of densified bamboo [16,63,64,65]. According to the microscopic results of Archila-Santos et al. (2016) [64], the vessels (vascular conduits) close as the result of densification. The breakdown of vacant spaces, such as protoxylem and phloem are also recognized (Figure 18, Figure 19 and Figure 20). Figure 20 shows the Scanning electron microscope analysis of the crosswise surface of non-densified and densified bamboo (*D. asper*) along with the specimens’ thickness and at different initial MC levels. The images show the closure of the cavities and the compaction through the cell wall thickness. It is also possible to see the different compaction of bamboo components such as fiber bundle and parenchyma. Kadivar et al. (2019) [65] stated that the densification happens principally in the central layer of the samples section [65].

### 7.6. Best Practices

The bamboo species, densification degree, temperature, and initial moisture content are the suggested effective parameters for the densification process. By combining the mentioned studies, there is an optimized range for each parameter, and using a parameter outside this range causes defects in the performance of the final product.

According to the available literature, although the best practice (in terms of bending and tensile resistance and also physical properties) depends on the bamboo species, it is possible to approximate the best water content as 7–10%, the best temperature between 140 and 160 °C, and densification degrees of 30–50% for the bamboo modification.

### 7.7. Gaps in Knowledge

There are some important gaps regarding other parameters such as pressure, time, and rate, which are also effective parameters that are scarcely informed in the publications. Based on the theory, the pressure rate is correlated with moisture content, since low rates can dry the material while high rates can also govern the incidence of cracks. On the other hand, in terms of the mentioned effective parameters, theoretically, there should be correlations that need to be identified. Identifying these correlations helps the process optimization, which will also lower the energy consumption and consequently reduce the associated costs and environmental impacts from manufacturing.

Existing studies have evaluated the process mostly in flexural strength and some physical properties. Information regarding the tensile properties of densified bamboo is scarce. Therefore, it is difficult to make a general conclusion about the mechanical properties of densified bamboo. Regarding some physical properties, such as spring back, swelling, and water absorption, at the best of the authors’ knowledge, little information is available, which accordingly, densification harms the dimensional stability [65]. Therefore, it can be an important topic for future investigations in this field to solve the problem.

Regarding the densification process, other methods that have been used for wood can be utilized for bamboo to solve some challenges, such as swelling and water absorption. Last but not least, using chemical modification as a pre-treatment can also facilitate the process.

## 8. Conclusions and Perspectives

In this review paper, the development of densified bamboo according to different processes recently introduced in the literature was discussed. Applying thermo-mechanical (TM), thermo-hydro-mechanical (THM), or viscoelastic-thermal-compression (VTC) methods allow the densification of bamboo specimens in the radial direction to approximately 20–67% of densification degree, increasing the density by 20–100%.

The densification process can decrease the heterogeneity and enhance bamboo mechanical performance. However, some physical properties, such as dimension stability, swelling, and water absorption, are reported to be compromised. Moreover, the process has an influence on chemical components of bamboo only at temperatures higher than 160 °C. The efficiency of densified bamboo can vary depending on the process parameters. Bamboo species, moisture content, hot-press temperature, pressing time, and pressure are the main factors that affect the densification process.

Though several aspects of these modifications are known, the fundamental influence of the process on the performance of densified bamboo has yet to be explored for the development of bamboo modification technologies. Further investigation is needed to design effective parameters for the processing and densification of bamboo.

## Figures and Tables

**Figure 1 materials-13-04346-f001:**
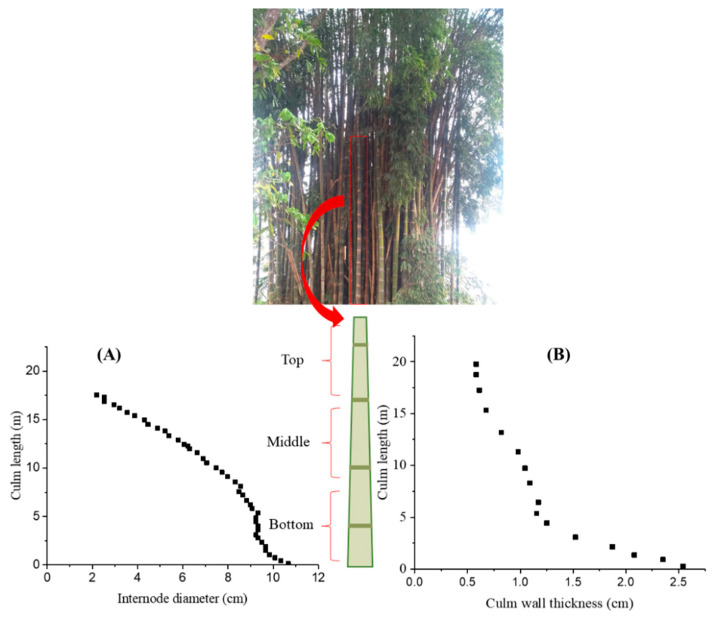
The variation of macroscopic characteristics along the culm length, an example of the bamboo species *Dendrocalamus asper*, (**A**) internode diameter and (**B**) culm wall thickness (data from [29]).

**Figure 2 materials-13-04346-f002:**
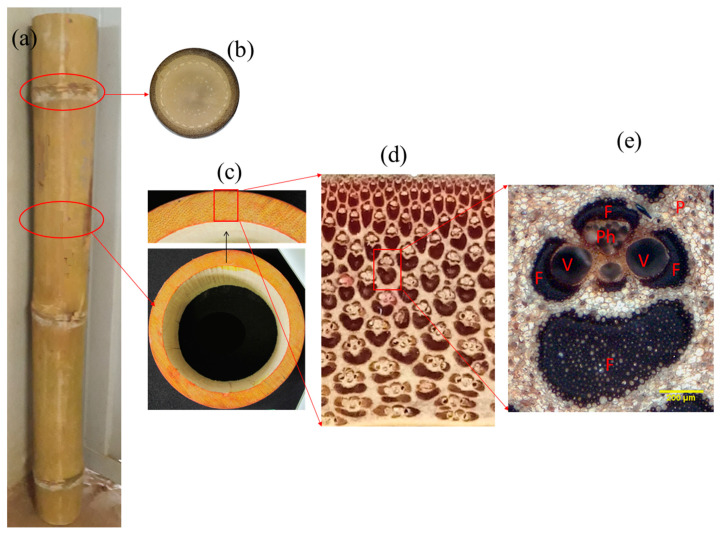
Overview of the morphological characteristic of bamboo culm. (**a**) Part of bamboo culm; (**b**) Cross section of bamboo showing node diaphragm; (**c**) Cross section of internode; (**d**) Section through culm wall; (**e**) Vascular bundle, V—Vessel, F—Fiber, Ph—Phloem, P—Parenchyma.

**Figure 3 materials-13-04346-f003:**
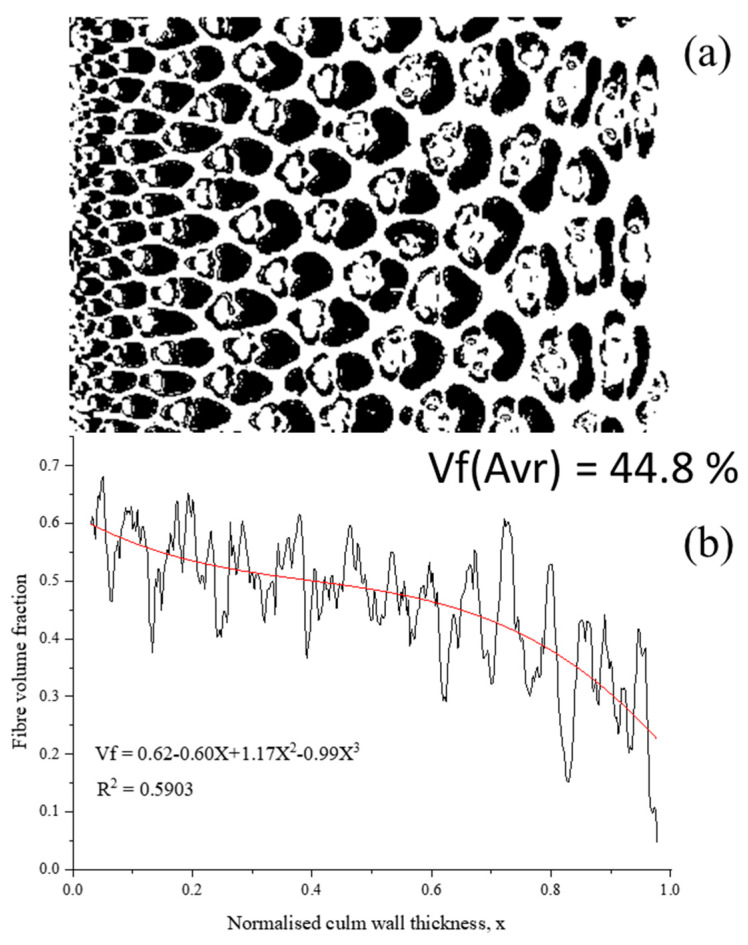
Fiber distribution in the composite bamboo (*D. Asper*). (**a**) example of digital image analysis; (**b**) fiber volume distribution across the culm wall.

**Figure 4 materials-13-04346-f004:**
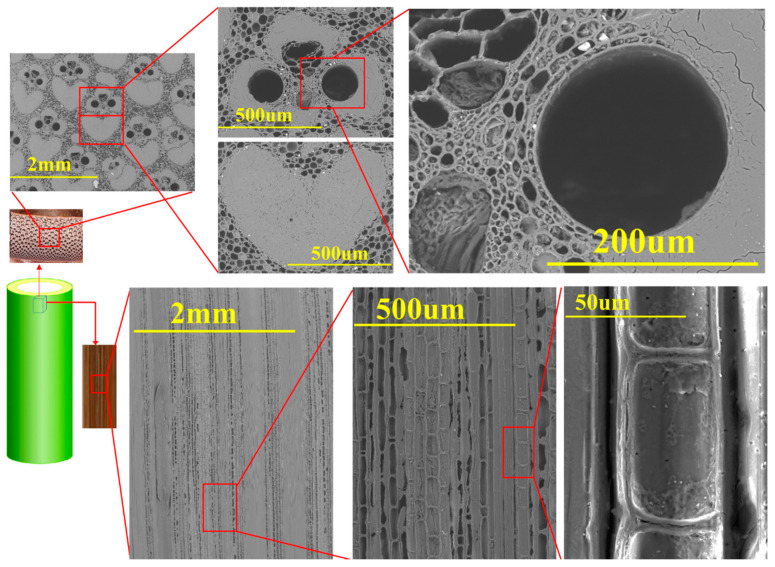
Details of the microstructure of the bamboo *D.asper*.

**Figure 5 materials-13-04346-f005:**
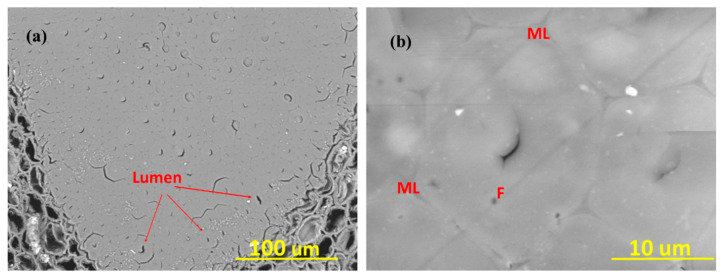
(**a**) Bamboo fiber bundle, (**b**) Elementary bamboo fibers (F) bonded by middle lamellae (ML).

**Figure 6 materials-13-04346-f006:**
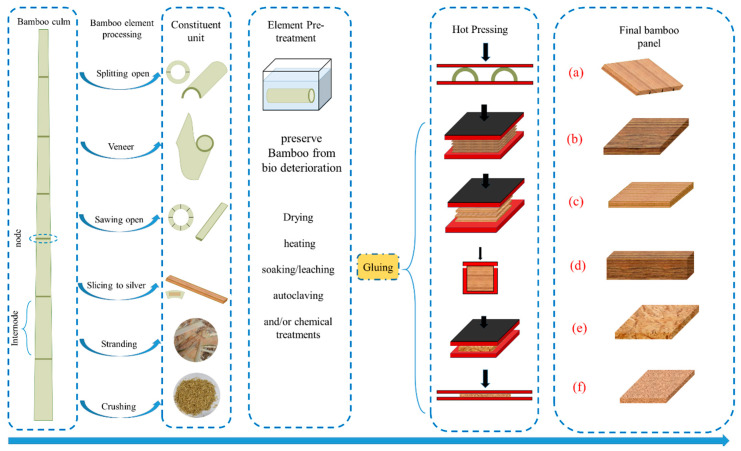
Summary of bamboo-based panel production processes. (**a**) Flattened bamboo panel, (**b**) Bamboo laminated lumber, (**c**) Plybamboo, (**d**) Bamboo scrimber, (**e**) Bamboo oriented strand board, (**f**) Bamboo particleboard).

**Figure 7 materials-13-04346-f007:**
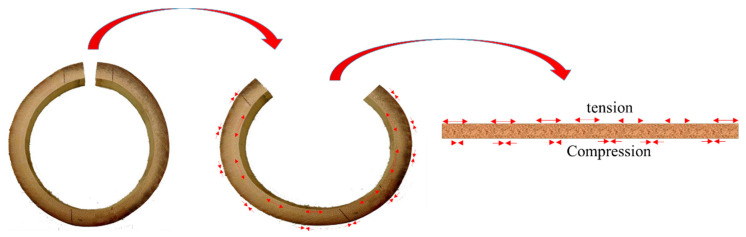
The mechanism of flattening.

**Figure 8 materials-13-04346-f008:**
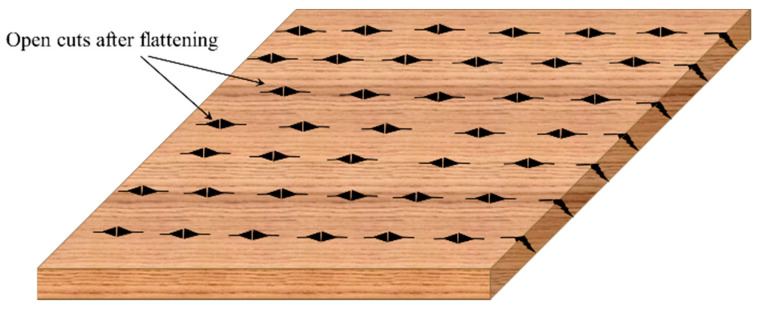
Scratches on the internal layer of flattened bamboo.

**Figure 9 materials-13-04346-f009:**
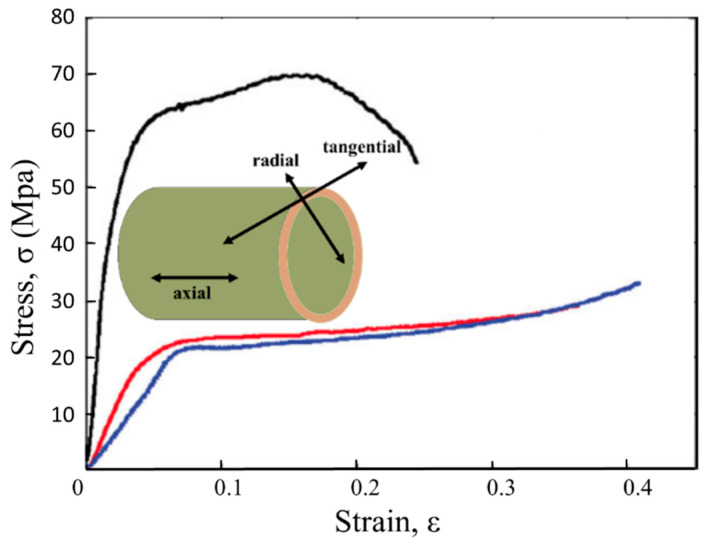
Typical compressive stress-strain curves of bamboo (adapted from [36]).

**Figure 10 materials-13-04346-f010:**
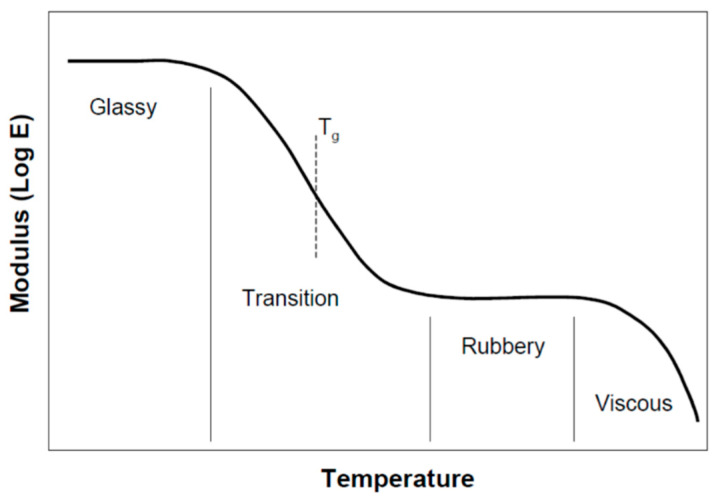
Variation of relaxation modulus with temperature for an amorphous polymer [88].

**Figure 11 materials-13-04346-f011:**
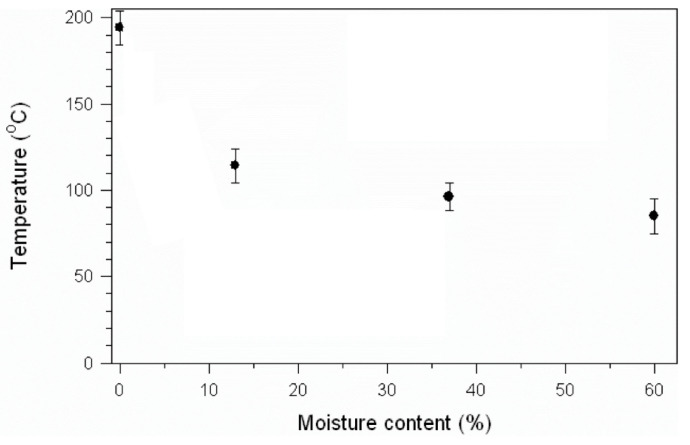
The moisture dependence of the glass transition (Tg) for bamboo (data from [82]).

**Figure 12 materials-13-04346-f012:**
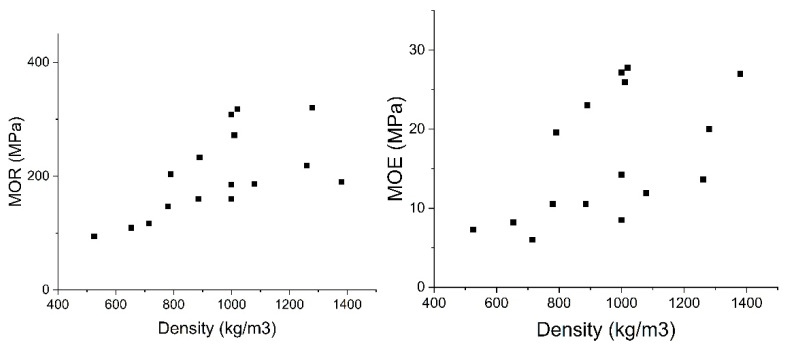
Bending-density results of data presented in Table 3.

**Figure 13 materials-13-04346-f013:**
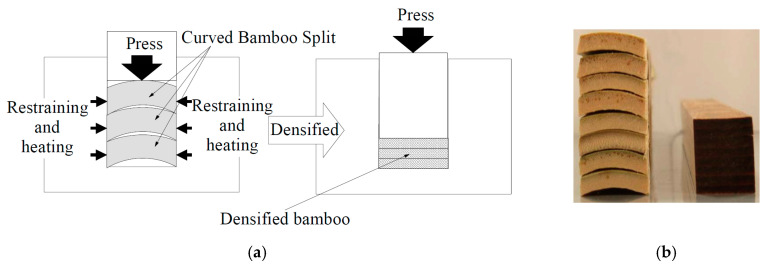
(**a**) Densification process used by Tanaka et al. (2006) [68] and (**b**) bamboo samples before and after the densification process [68].

**Figure 14 materials-13-04346-f014:**
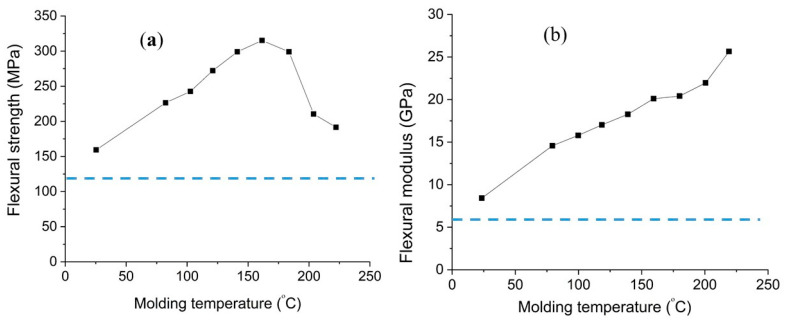
(**a**) Flexural strength, and (**b**) modulus as a function of molding temperature (the dashed line shows the flexural strength and modulus of un-densified bamboo), data from [67].

**Figure 15 materials-13-04346-f015:**
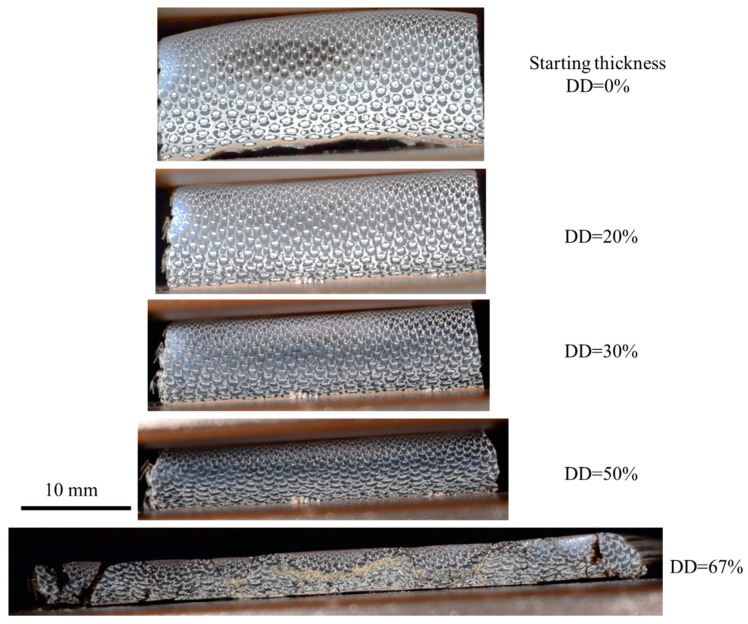
Cross sections of D.Asper strips densified at 160 °C.

**Figure 16 materials-13-04346-f016:**
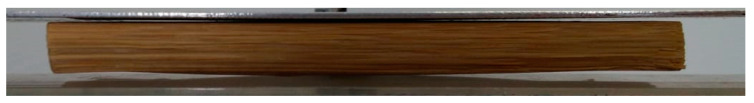
Bamboo sample densified with 20% MC.

**Figure 17 materials-13-04346-f017:**
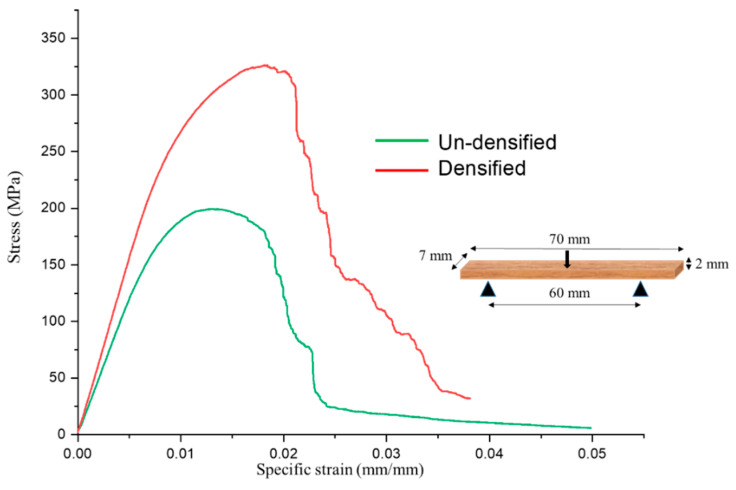
Stress-strain bending plot of non-densified and densified samples with 10% of moisture content (data from [65]).

**Figure 18 materials-13-04346-f018:**
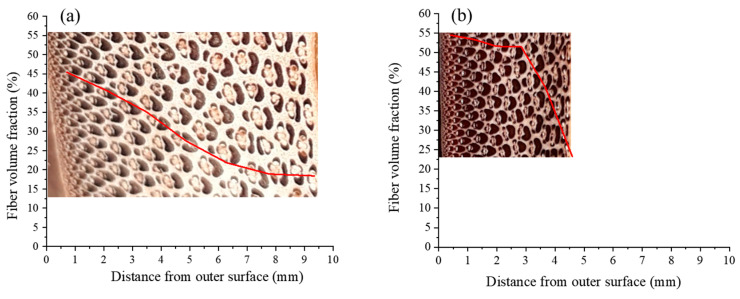
Fiber volume fraction of (**a**) normal bamboo, and (**b**) reformed bamboo (data from [31]).

**Figure 19 materials-13-04346-f019:**
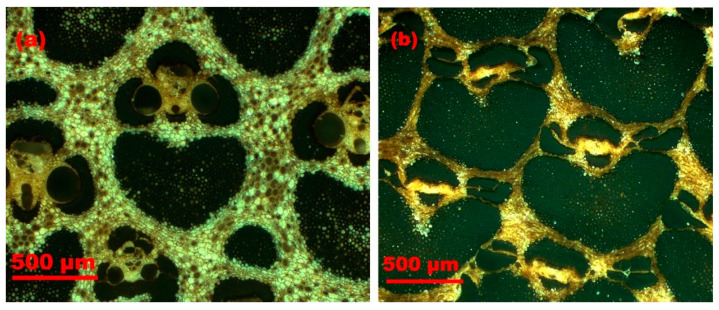
Optical microscope images (**a**) before densification and (**b**) after densification.

**Figure 20 materials-13-04346-f020:**
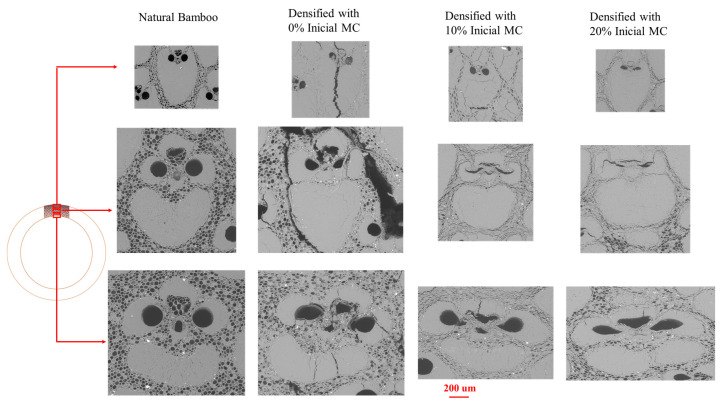
Scanning electron microscope (SEM) images of the non-densified and densified samples in different initial MC (from 0% up to 20%).

**Table 1 materials-13-04346-t001:** The classification of bamboo elements used in densified products (specifically bamboo based panels (BBPs)).

Bamboo Element	Production Method	Size Range (mm)	Description	References
Half-split Culm (Half-Round Bamboo) 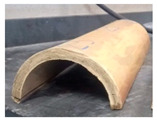	Dividing a culm into two equal splits and removing the nodes.	The size of half-split culm depends on the original culm dimension	It has been used for a variety of traditional applications such as roof tiles and drainage ducts.	[41,50]
Bamboo Veneer [51] 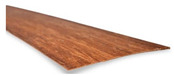	Rotary cutting of bamboo culms	T = 0.15–1.5W = variedL = varied	A sheet-like folio which has the biggest bamboo industrial element size.	[39,50,51,52]
Bamboo Strip 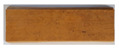	Cutting bamboo culm in longitudinal direction or by flattening half splits	T = 3–10W = 15–25L = varied	It is a long, thin piece of culm with uniform thickness and width.	[39,41,53,54]
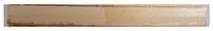 Bamboo Sliver	Cutting processSlicing to sliver	T = 0.5–3.5W = 10–30L = varied	Are also long bamboo elements consisting of flat surfaces, thinner than strips.	[27,29]
Bamboo Strands 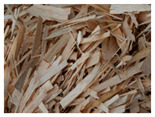	Produced by disk flaker	T: 0.3–0.8W: 5–20L: 50–90	Are consisted of flat, long bundles of fibers having parallel surfaces and short slivers of bamboo.	[29,39,47,49,55,56]
Bamboo Particles 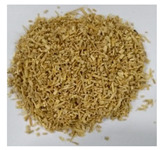	Milling	T: 0.1–0.5W: 1–5L: 20–30		[39,57,58]

**Table 2 materials-13-04346-t002:** Bamboo densification process parameters.

Ref.	Bamboo Species	Size of Samples (mm)	Temperature (°C)	Pressing Time (min)	Method of Press	Pressure MPa	Densification Degree (DD) (%)
[68]	*Phyllostachys edulis*	I + M + O	120	-	Thermo-hydro-mechanical (THM)	8	-
	*P. bambusoides*	I + M + O					
[67]	*P. bambusoides*	10 × 200 I + M + O	25–220	10	Thermo-mechanical (TM)	50	-
[94]	*P. edulis*	5 × 20 × 150 M + I	170	13.3	Viscoelastic-thermal-compression (VTC) (steam pressure 775 kPa)		20–67
[64]	*Guadua angustifolia*	M	150	20	TM	6.2	42.51
[16]	*P. edulis*	90 ×20 × (3–5)	170	13.3	VTC (steam pressure 775 kPa)	-	50
[65]	*Dendrocalamus asper*	I+M+O	140	20	TM	4.34	31.2

I: inner layer, M: middle layer, O: outer layer, DD: Densification Degree.

**Table 3 materials-13-04346-t003:** Summary of the mechanical and physical properties of bamboo, before and after densification process.

Reference	Variable	Standard Method or Specimen Dimension	Apparent Density (kg/m^−3^)		Moisture Content (MC) (%)		Modulus of Elasticity (MOE) (GPa)		Ultimate Stress (MPa)	
			Before	After	Before	After	Before	After	Before	After
[68]	*P. edulis*	JIS Z2101. (1994) [95]	680	1334	8.5–36.5	-	12.0 (T)	32.0 (T)	80 (T)	310 (T)
	*P. bambusoides*		600	1100		-	8.5 (T)	19.0 (T)	170 (T)	220 (T)
[67] *(P. bambusoides)*	25 °C	(10 × 100 × (3–5)) mm	714	1000	5 ± 3	-	6 (B)	8.5 (B)	117 (B)	160 (B)
	160 °C			1280				20.0 (B)		320 (B)
	220 °C			1380				27.0 (B)		190 (B)
[94] *(P. edulis)*	80% C. R	ASTM D1037-06a (2006) [96]	653.9	780	9	6.8	8.21 (B)	10.5 (B)	109 (B)	147 (B)
	66% C. R			885		6.7		10.5 (B)		160 (B)
	50% C. R			1079		7.7		11.9 (B)		187 (B)
	33% C. R			1261		7.7		13.6 (B)		219 (B)
[64] *(G. angustifolia)*	dry	ISO 22157 (2004) [97,98]	540 (Oven dried)	810 (OD)	-	-	16.2 (T)	22.8 (T)	-	-
	Pre-soaked			830 (OD)	-	-		31.0 (T)		-
[16] *(P. edulis)*	-	70 × 5 × (1–3) mm	450 to 600	800 to 1200	7	5	2.5–12.1 (B)	5.4–23.0 (B)	47–140 (B)	74–296 (B)
[65] *(D.Asper)*	0% MC	ASTM D7264—15 (2015) [99]	790	890	0	-	19.6 (B)	23.0 (B)	203 (B)	233 (B)
	5% MC			1000	5			27.1 (B)		308 (B)
	10% MC			1020	10			27.8 (B)		318 (B)
	20% MC			1010	20			25.9 (B)		272 (B)

T: Test in tensile, B: Test in bending, OD: Oven dried, JIS Z2101: Japanese Standards Association, ASTM: American Society for Testing and Materials International, ISO: International Organization for Standardization.

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
