# Peer review of "Densification of Bamboo: State of the Art"

_materials, 2020, doi:10.3390/ma13194346_

Round 1

Reviewer 1 Report

Densification of bamboo: state of art,

The publication deals with linear and flat materials made of bamboo. The processes always involve humidity and temperature and force, so that one can generally speak of THM processes. Compaction is only one specific process among others. Even if it is the main focus of attention, there is also the flattening, which can be equated more with shaping. Against this background, a new formulation of the title can again be considered, although the title is very short and concise.

Line 80 The reference (17) does not apply to wood composites, but to bamboo

Line 94 same counts for this reference (24)

General comment: Kutnar et al. published a recent review of THM treatments in the journal Holzforschung 2015, Compressed and moulded wood from processing to products - a review. This paper summarizes the state of the art and knowledge in this field. This more recent paper of Kutnar et al. should be mentioned in such a comprehensive literature review.

Line 103 "The internodes are cylinders ..." This is a strange sentence, which explains the term to be defined with the same term ... please change!

Line 126 Are you sure that homogeneity is the decisive disadvantage of bamboo in the building sector. What about the variations of the small diameters, which means an enormous effort in connection with the short culm. And what about the cracking caused by stiff internodes during shrinking?

Figure 4: show the inside and outside of the cross-section on the figure

Figre 11: Where to see the resin? Black or white color in the lumen? Use arrows

Line 243 What is the purpose of compacting bamboo in construction? For tensile loads it makes no sense. For bending and buckling you lose moments of inertia. The only advantage is the smaller size of the cross section, which is not crucial in building construction but perhaps in other technical fields.

Figure 13: This figure corresponds to the state of the art. It has no direct relation to bamboo and should be well known to the reader. Leave it away.

Line 299 Do you use springback and shape memory as synonyms? As I understand it, springback occurs when the sample is compressed and immediately discharged, shape memory or recovery when a fixed compressed sample is moistened and/or heated. Is that what you want to say?

In the paper there is no mention of shape recovery of compacted bamboo. Does it not exist?

Line 327 write consistently viscoelastic-thermal compression thermo-mechanical ... with small letters in the text and use the abbreviation VTC, THM with capital letters except for the headings.

Line 382 Bamboo Connector means Bamboo Connection or Connection?

Line 388 How do you explain the different relative improvement? Is it due to damage?

Line 514 Apparently, the strength does not increase proportionally with density. Is there an explanation for this?

Figure 20 The stress strain behavior during bending seems to be very ductile even for compacted bamboo. Such a behavior is not to be expected. Wh

Author Response

We are grateful to all of your suggestions and corrections, which were very important in improving the quality of this manuscript.

Point 1: The publication deals with linear and flat materials made of bamboo. The processes always involve humidity and temperature and force, so that one can generally speak of THM processes. Compaction is only one specific process among others. Even if it is the main focus of attention, there is also the flattening, which can be equated more with shaping. Against this background, a new formulation of the title can again be considered, although the title is very short and concise.

Response 1: Thank you for the comment. You are correct, however, in the manuscript the focus is on the pure densification. As it is explained in the section 4.1. of the paper, we separated the flattening from densification because it involves different mechanisms.

Point 2: Line 80 The reference (17) does not apply to wood composites, but to bamboo

Response 2: The reference has been corrected.

Point 3: Line 94 same counts for this reference (24)

Response 3: The reference has been corrected.

Point 4: General comment: Kutnar et al. published a recent review of THM treatments in the journal Holzforschung 2015, Compressed and moulded wood from processing to products - a review. This paper summarizes the state of the art and knowledge in this field. This more recent paper of Kutnar et al. should be mentioned in such a comprehensive literature review.

Response 4: Thank you for this suggestion. We have included the mentioned reference in the line 56, reference number 14.

Point 5: Line 103 "The internodes are cylinders ..." This is a strange sentence, which explains the term to be defined with the same term ... please change!

Response 5: The sentence has been corrected: “The internodes have a hollow cylindrical shape, while the nodes are thin diaphragms separating the internodes”

Point 6: Line 126 Are you sure that homogeneity is the decisive disadvantage of bamboo in the building sector. What about the variations of the small diameters, which means an enormous effort in connection with the short culm. And what about the cracking caused by stiff internodes during shrinking?

Response 6: Defenitelythe variations of the small diameters and the cracking caused by stiff internodes during shrinking, are also other disadvantages of bamboo in the building sector. Therefore, line 126 has been modified to “ The lack of homogeneity across the bamboo thickness is one of the negative points of using bamboo culm directly in the construction industry.

Point 7: Figure 4: show the inside and outside of the cross-section on the figure

Response 7: The figure has been  corrected

Point 8: Figre 11: Where to see the resin? Black or white color in the lumen? Use arrows

Response 8: The figure has beens corrected

Point 9: Line 243 What is the purpose of compacting bamboo in construction? For tensile loads it makes no sense. For bending and buckling you lose moments of inertia. The only advantage is the smaller size of the cross section, which is not crucial in building construction but perhaps in other technical fields.

Response 9: In the construction industry, due to the use of bamboo panels, bamboo is compacted in different processes. In this article, this densification process is investigated in a focused way. Regarding the bamboo strength, we completely agree with you. However, as you can see in table 3, in some cases, densified bamboo has a tensile strength around 3 times higher than the natural bamboo. In addition, as you mentioned, the application can be in other technical fields as well.   

Point 10: Figure 13: This figure corresponds to the state of the art. It has no direct relation to bamboo and should be well known to the reader. Leave it away.

Response 10: We understood your comment. However, bamboo undergoes the same phases by increasing the temperature.

Point 11: Line 299 Do you use springback and shape memory as synonyms? As I understand it, springback occurs when the sample is compressed and immediately discharged, shape memory or recovery when a fixed compressed sample is moistened and/or heated. Is that what you want to say?

Response 11: The sentence has been adjusted.  

Point 12: In the paper there is no mention of shape recovery of compacted bamboo. Does it not exist?

Response 12: We have used springback and shape recovery as synonyms. We could only find little  information related to shape recovery of compacted bamboo that we mentioned in the menuscript “However, in terms of physical properties and dimensional stability, the densified samples in all samples resulted in decreased performance compared to the natural bamboo”. Since this is an important feature of densified bamboo, we have included that in the gap of knowledge (line 531) “Regarding some physical properties, such as spring back, swelling, and water absorption, at the best of the authors’ knowledge, little information is available, which accordingly, densification harms the dimensional stability [59]. Therefore, it can be an important topic for future investigations in this field to solve the problem. “

Point 13: Line 327 write consistently viscoelastic-thermal compression thermo-mechanical ... with small letters in the text and use the abbreviation VTC, THM with capital letters except for the headings.

Response 13: Agreed and corrected

Point 14: Line 382 Bamboo Connector means Bamboo Connection or Connection?

Response 14: More explanation added, Tanaka et al. produced the bamboo connector for developing a new connecting system”.

Point 15: Line 388 How do you explain the different relative improvement? Is it due to damage?

Response 15: Different fiber content can be a reason to have different results. This justification has been added to the text.

Point 16: Line 514 Apparently, the strength does not increase proportionally with density. Is there an explanation for this?

 Response 16: In this line, the comparison is between densification degree (The change of the bamboo thickness) and density using different methods.

Point 17: Figure 20 The stress strain behavior during bending seems to be very ductile even for compacted bamboo. Such a behavior is not to be expected. Wh

Response 17: This behaviour might be because of the size of the samples, which were very small (2*7*70 mm). These samples were tested up to a specific deflection (0.05) and not only up to maximum stress.  Additionally, the bending behaviour of bamboo is different from wood. After the limit of proportionality, this “plastic” region is characterized by fiber detachment and failure. Other authors also observe the same bahavior. Check: “Gibson, L.J., Dixon, P.G., 2014. The structure and mechanics of Moso bamboo material. J. R. Soc. Interface 11. https://doi.org/10.1098/rsif.2014.0321”

Reviewer 2 Report

Overall, this is a very nicely written review paper on the densification of bamboo.  I have only a few minor suggestions to improve the manuscript, which I've listed below.

  1. Lines 95-97: I suggest to add some detail about what is lacking in the literature, what are the gaps that future research need to address.
  2. Figure 2: Please improve the resolution of this figure, particularly the bar plot. Also, please add in the figure caption a description of the numbers in parenthesis (I presume they are standard errors or deviations of the average contact angle).
  3. Line 118: I suggest to replace "fibers bundles" for "fiber bundles" or "bundles of fibers".
  4. Figure 3 (a): I suggest to add an arrow to point the node.
  5. Line 140:  Could you add an estimate of the required force, perhaps based on the cell wall mechanical properties.  A comparison between the tangential and radial orientation may be useful to the reader.
  6. Line 262:  Is the lignin Tg the limiting factor?  I suggest adding an explanation of why pre-treatments and/or using moisture to decrease the Tg of lignin would be beneficial for geometric deformation.  Another option would be to not go into detail here, since the viscoelastic and softening of bamboo is discussed in more detail in the following sections.
  7. Line 300: Please elaborate, what is an appropriate method?
  8. Figure 17: What is the dashed line?  Please add a label or a description in the figure caption.
  9. Table 3: What is before and after?  I presume is before and after densification.  Please add the information.
  10. Figure 22: Are the lines fits to the data or only meant to be "guides"? Please add a description in the caption.
  11. Author contribution is blank.  Please verify that there is no missing text in the manuscript.

Author Response

We are grateful to all of your suggestions and corrections, which were very important in improving the quality of this manuscript. 

Point 1: Lines 95-97: I suggest to add some detail about what is lacking in the literature, what are the gaps that future research need to address.

Response 1: Thank you for the comments. Since there were several gaps, we allocated a separate section as 7.7. Gaps in knowledge”.

Point 2: Figure 2: Please improve the resolution of this figure, particularly the bar plot. Also, please add in the figure caption a description of the numbers in parenthesis (I presume they are standard errors or deviations of the average contact angle).

Response 2: We are asking the author for the permission of using this photo and we are asking for the high-resolution image. Before the final decision of submitting this paper, we will change the image to a high-resolution one.

Point 3: Line 118: I suggest to replace "fibers bundles" for "fiber bundles" or "bundles of fibers".

Response 3: Agreed and corrected to "fiber bundles"

Point 4: Figure 3 (a): I suggest to add an arrow to point the node.

Response 4: Agreed and edited.

Point 5: Line 140:  Could you add an estimate of the required force, perhaps based on the cell wall mechanical properties.  A comparison between the tangential and radial orientation may be useful to the reader.

Response 5: The required force varies by the species. In Figure 12, it is presented the typical compressive stress-strain curves of bamboo in axial, tangential, and radial orientations for Moso species.

Point 6: Line 262:  Is the lignin Tg the limiting factor?  I suggest adding an explanation of why pre-treatments and/or using moisture to decrease the Tg of lignin would be beneficial for geometric deformation.  Another option would be to not go into detail here, since the viscoelastic and softening of bamboo is discussed in more detail in the following sections.

Response 6: We mean the overall Tg of the material. The related sentence in this section has been deleted.

Point 7: Line 300: Please elaborate, what is an appropriate method?

Response 7: The sentence “An appropriate method is a process in which the optimized parameters apply to the material” has been added to the text, line 319.

Point 8: Figure 17: What is the dashed line?  Please add a label or a description in the figure caption.

Response 8: The dashed line in figure 17 shows the flexural strength and modulus of un-densified bamboo (the description has been added to the text).

Point 9: Table 3: What is before and after?  I presume is before and after densification.  Please add the information.

Response 9: Agreed and the title of the table has been modified to “Table 3. Summary of the mechanical and physical properties of bamboo, before and after densification process”.

 Point 10: Figure 22, Are the lines fits to the data or only meant to be "guides"? Please add a description in the caption.

Response 10: In the related reference, the authors  did not mention anything about this.

Point 11: Author contribution is blank.  Please verify that there is no missing text in the manuscript.

Response 11: Regarding the Author contributions section, this sentence has been added; “All of the authors contributed equally to this work”.

Reviewer 3 Report

This paper addressed an important topic in bamboo product manufacturing and product performance by assembling results from a large number of relevant papers. While information presented was useful in general, there is a lack of scientific depth.  The manuscript is more like a report than a scientific paper.  For example, Densification Concept (line 69 and the entire section) was explained in a way that was superficial (e.g. eliminating pores/cavities leads to densification, because of obviousness) and improper (e.g. density increases with resin impregnation, because it is actually a formation of new composite material).  The mechanics of perpendicular-to-grain wood or bamboo compression was not described sufficiently either (see Gibson and Ashby's book: Cellular Solids).

The lack of knowledge in wood and bamboo composites manufacturing seemed also apparent as no mentioning of the obvious relation of densification to developing intimate interfacial contact for bonding.  For example,

  • Densification degree (DD, line 258) could be replaced by more commonly used term such as compression or compaction ratio (CR). 
  • Line 419, "...a partial side spread of the tissue" should be properly described as "...lateral expansion".  
  • Line 443: "...release internal water" should be "...internal gas pressure".  Degasing or venting is very common in hot pressing of wood/bamboo composites.  
  • Line 239-242: "full-culm densification" or "just densification of bamboo wall"?  How could one laminate bamboo product without adhesives?

Author Response

We are grateful to all of your suggestions and corrections, which were very important in improving the quality of this manuscript. 

Point 1: This paper addressed an important topic in bamboo product manufacturing and product performance by assembling results from a large number of relevant papers. While information presented was useful in general, there is a lack of scientific depth.  The manuscript is more like a report than a scientific paper.  For example, Densification Concept (line 69 and the entire section) was explained in a way that was superficial (e.g. eliminating pores/cavities leads to densification, because of obviousness) and improper (e.g. density increases with resin impregnation, because it is actually a formation of new composite material).  The mechanics of perpendicular-to-grain wood or bamboo compression was not described sufficiently either (see Gibson and Ashby's book: Cellular Solids).

Response 1: Thank you for your comment. We have tried to improve this section. The mentioned reference and also another paper of Gibson related to this topic have been added to the manuscript.

Point 2: The lack of knowledge in wood and bamboo composites manufacturing seemed also apparent as no mentioning of the obvious relation of densification to developing intimate interfacial contact for bonding.  For example,

Densification degree (DD, line 258) could be replaced by more commonly used term such as compression or compaction ratio (CR). 

Response 2: The term DD is used in several papers that studied the densification process of bamboo and wood.

Point 3: Line 419, "...a partial side spread of the tissue" should be properly described as "...lateral expansion".  

Response 3: Agreed and added to the manuscript.

Point 4: Line 443: "...release internal water" should be "...internal gas pressure".  Degasing or venting is very common in hot pressing of wood/bamboo composites.  

Response 4: Agreed and corrected

Point 5: Line 239-242: "full-culm densification" or "just densification of bamboo wall"?  How could one laminate bamboo product without adhesives?

Response 5: Some description has been added in line 254 to make the concept more clear: “(or strips with whole thickness of bamboo)”.

Reviewer 4 Report

I find the manuscript entitled  ”Densification of Bamboo: State of the Art” interesting.  I am sure that it will be useful for researchers working on the development of new materials based on poor-quality fast-growing wood species. My sincere congratulations to the Authors for their excellent work.

In my opinion, only some editorial corrections are required:

  • in the “2. Densification Concept” chapter – when we talk about the process which involves high temperature, it is necessary to mention not only the changes in wood structure but also in wood chemical composition, which affects wood properties;
  • the “References” part definitely requires a thorough verification. There is a lot of titles or names in capitals, some journal names or author’s names are missing etc. (e.g. No 81 or 82).

Author Response

We are grateful to all of your suggestions and corrections, which were very important in improving the quality of this manuscript. Please find attached the response letter point-by-point related to the comments. 

Round 2

Reviewer 3 Report

I am glad to see the effort to modify the manuscript.  This should be useful addition to the literature about bamboo products.